# ADAPTIVEMIXGNN: LOCAL ADAPTIVE INDUCTIVE BIAS FOR HETEROPHILIC NODE CLASSIFICATION

**Miguel Alcocer Pérez, Javier Muñoz de Torres & Álvaro Morán Lorente**
Universidad Rey Juan Carlos, Madrid, Spain
{m.alcocer.2022,j.munozd.2022,a.moranl.2022}@alumnos.urjc.es

## ABSTRACT

Most GNNs apply a *uniform* global filter to every node, implicitly assuming one dominant structural regime. Real graphs violate this assumption: homophilic and heterophilic patterns coexist and vary *locally*. We introduce **AdaptiveMixGNN**, a first-order spectral GNN that preserves *scale and simplicity* by learning a per-node mixing between low-pass and high-pass shifts, $\mathbf{S}_{\boldsymbol{\alpha}} = \mathrm{diag}(\boldsymbol{\alpha})\,\mathbf{S}_{\mathrm{LP}} + (\mathbf{I} - \mathrm{diag}(\boldsymbol{\alpha}))\,\mathbf{S}_{\mathrm{HP}}$, with $\alpha_i = \sigma(\mathbf{h}_i^\top \boldsymbol{\theta} + b)$. This adds only $d{+}1$ parameters per layer and keeps $O(L{\cdot}|\mathcal{E}|)$ complexity (GCN-like). On heterophilic benchmarks, AdaptiveMixGNN reaches **79.46%** on Texas and **79.61%** on Wisconsin, outperforming polynomial filters we evaluated ($K{\geq}10$) while avoiding their overfitting pathologies on small graphs. Ablations show that node-wise adaptivity acts as an *insurance policy* against catastrophic failures of fixed filters, with gains up to **+10.59%** over the best static baseline. Finally, a per-node homophily analysis links the learned $\boldsymbol{\alpha}$ to local label structure (Texas: $\bar{h}{=}0.033$ vs. $0.247$ for correct vs. incorrect nodes), suggesting that the model discovers a meaningful local frequency response.

## 1 INTRODUCTION

Message-passing GNNs act as low-pass graph filters, smoothing features along edges (Zhu et al., 2020). This inductive bias succeeds under homophily but fails when neighbours carry *different* labels. Two families of solutions dominate the literature: (i) polynomial spectral filters of order $K$ (GPRGNN (Chien et al., 2021), BernNet (He et al., 2021), JacobiConv (Wang & Zhang, 2022)) that learn global frequency responses at $O(K{\cdot}|\mathcal{E}|)$ cost; and (ii) heterophily-specific architectures (H2GCN (Zhu et al., 2020), ACM-GCN++ (Luan et al., 2022)) that redesign aggregation. Both apply the *same* filter uniformly to all nodes. FAGCN (Bo et al., 2021) partially relaxes this by learning per-*edge* signed attention weights that blend low- and high-frequency components; however, it still requires computing attention over all edges and does not expose a per-node gate derived from node features alone.

Han et al. (2025) prove that a global filter optimized for one structural pattern can *damage* performance on nodes with the opposite pattern (Theorem 1 therein). Their Node-MoE solution deploys multiple full-scale expert GNNs. We pursue a radically simpler path—motivated by *scale and simplicity*—that stays first-order and linear-time: a **local adaptive inductive bias** that lets each node select its own frequency response from a *minimal* two-component filter bank, at negligible parameter and computational overhead.

**Contributions.** (1) We propose AdaptiveMixGNN, a first-order spectral GNN with node-wise adaptive mixing between low-pass ($\mathbf{S}_{\mathrm{LP}}$) and high-pass ($\mathbf{S}_{\mathrm{HP}}$) operators. (2) We prove permutation equivariance, anchoring the architecture in the symmetry framework of geometric deep learning. (3) We achieve state-of-the-art among $O(|\mathcal{E}|)$ methods on Texas and Wisconsin while maintaining competitive homophilic performance. (4) Ablation and per-node homophily analyses provide geometric grounding for the learned $\alpha_i$ values.

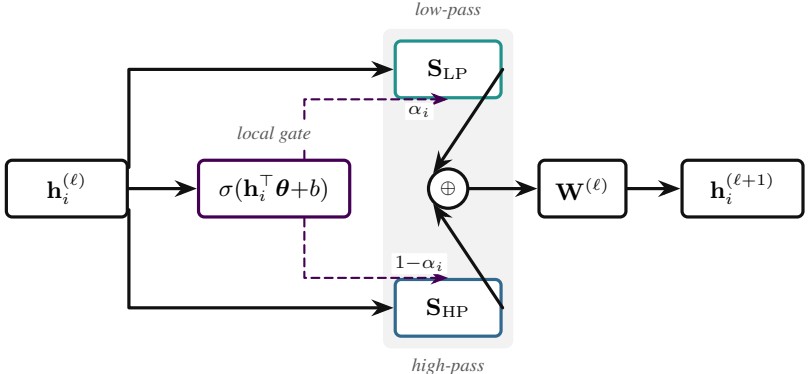

Figure 1: One AdaptiveMixGNN layer (node-centric view). A local gate $\alpha_i$ blends low-/high-pass shifts before the shared linear map, preserving first-order $O(|\mathcal{E}|)$ scaling.

## 2 METHOD

### 2.1 PRELIMINARIES AND NOTATION

Let $\mathcal{G} = (\mathcal{V}, \mathcal{E})$ with $|\mathcal{V}| = n$, $|\mathcal{E}| = m$, adjacency $\mathbf{A}$, and features $\mathbf{X} \in \mathbb{R}^{n \times d}$, where $x_i^\top$ denotes the $i$-th row of $\mathbf{X}$. Following Han et al. (2025), define the normalised adjacency $\tilde{\mathbf{A}}_{\mathrm{sym}} = \tilde{\mathbf{D}}^{-1/2} \tilde{\mathbf{A}} \tilde{\mathbf{D}}^{-1/2}$ ($\tilde{\mathbf{A}} = \mathbf{A} + \mathbf{I}$). The node homophily $h(v_i) = |\{u \in \mathcal{N}(v_i) : y_u = y_i\}|/d_i$ is a standard local label-agreement statistic used in prior heterophily evaluations (Platonov et al., 2023). In the spectral domain, $\tilde{\mathbf{A}}_{\mathrm{sym}}\mathbf{X}$ applies filter $f(\lambda) = 1 - \lambda$ (low-pass); $(\mathbf{I} - \tilde{\mathbf{A}}_{\mathrm{sym}})\mathbf{X}$ applies $f(\lambda) = \lambda$ (high-pass).

### 2.2 ARCHITECTURE

We define the **low-pass** and **high-pass** graph shift operators (GSOs):
$$\mathbf{S}_{\mathrm{LP}} = \tilde{\mathbf{D}}^{-1/2} \tilde{\mathbf{A}} \tilde{\mathbf{D}}^{-1/2}, \qquad \mathbf{S}_{\mathrm{HP}} = \mathbf{I} - \mathbf{S}_{\mathrm{LP}} \tag{1}$$
These two operators span the space of first-order graph polynomials $c_0 \mathbf{I} + c_1 \tilde{\mathbf{A}}_{\mathrm{sym}}$, forming a *minimal basis*: no smaller set can represent all first-order spectral filters.

Rather than learning a single global interpolation, AdaptiveMixGNN computes a **node-wise mixing coefficient per layer** from the current node representation $\mathbf{h}_i^{(\ell)}$:
$$\alpha_i^{(\ell)} = \sigma\left( \left( \mathbf{h}_i^{(\ell)} \right)^\top \boldsymbol{\theta}^{(\ell)} + b^{(\ell)} \right), \quad \boldsymbol{\theta}^{(\ell)} \in \mathbb{R}^{d_\ell}, \; b^{(\ell)} \in \mathbb{R} \tag{2}$$
yielding the adaptive shift operator and layer update:
$$\mathbf{S}_{\boldsymbol{\alpha}} = \mathrm{diag}(\boldsymbol{\alpha}) \, \mathbf{S}_{\mathrm{LP}} + (\mathbf{I} - \mathrm{diag}(\boldsymbol{\alpha})) \, \mathbf{S}_{\mathrm{HP}} \tag{3}$$
$$\mathbf{H}^{(\ell+1)} = \sigma_{\mathrm{act}} \left( \mathbf{S}_{\boldsymbol{\alpha}^{(\ell)}} \, \mathbf{H}^{(\ell)} \, \mathbf{W}^{(\ell)} \right) \tag{4}$$
where $\mathbf{H}^{(0)} = \mathbf{X}$. Figure 1 illustrates one layer.

### 2.3 COMPLEXITY: FIRST-ORDER BY DESIGN

Each layer applies $\mathbf{S}_{\mathrm{LP}}\mathbf{H}$ via one sparse matrix–matrix product ($O(md)$); since $\mathbf{S}_{\mathrm{HP}} = \mathbf{I} - \mathbf{S}_{\mathrm{LP}}$, the high-pass branch can be computed as $\mathbf{S}_{\mathrm{HP}}\mathbf{H} = \mathbf{H} - \mathbf{S}_{\mathrm{LP}}\mathbf{H}$ with an additional $O(nd)$ subtraction. The remaining costs are one element-wise gating ($O(nd)$) and one dense projection ($O(nd^2)$). For constant feature dimension $d$, the per-layer cost is $O(m)$—**identical to GCN**. By contrast, polynomial filters of order $K$ (GPRGNN, JacobiConv, BernNet, ChebNet) require $K$ successive sparse multiplications, yielding $O(Km)$. With typical $K \geq 10$, this represents an order-of-magnitude overhead. As He et al. (2022) demonstrate, this additional capacity does not translate into accuracy gains on small graphs; instead, it causes overfitting via "illegal" polynomial coefficients.

**Parameter overhead.** Each AdaptiveMixGNN layer adds exactly $d+1$ parameters for $(\boldsymbol{\theta}, b)$—the minimal cost of local adaptivity.

Table 1: Node classification accuracy (%). **Bold**: best; underline: second. $\Delta_{\mathrm{mix}}$: gain of adaptive $\alpha$ over best fixed filter.

| Model | Cost | Cora $(h{=}.81)$ | Texas $(h{=}.11)$ | Wisc. $(h{=}.21)$ | Cham. $(h{=}.23)$ | Actor $(h{=}.22)$ |
|---|---|---|---|---|---|---|
| MLP | $O(n)$ | $57.1_{\pm1.1}$ | $\mathbf{78.9}_{\pm2.0}$ | $\underline{78.4}_{\pm2.5}$ | $47.9_{\pm1.3}$ | $35.2_{\pm0.6}$ |
| GCN | $O(m)$ | $81.0_{\pm0.9}$ | $62.2_{\pm2.1}$ | $55.7_{\pm2.4}$ | $39.9_{\pm1.9}$ | $28.7_{\pm0.6}$ |
| GAT | $O(m)$ | $80.4_{\pm1.3}$ | $56.5_{\pm10}$ | $47.7_{\pm6.4}$ | $47.0_{\pm2.0}$ | $29.1_{\pm0.7}$ |
| H2GCN | $O(m)$ | $80.6_{\pm0.6}$ | $73.0_{\pm4.4}$ | $69.6_{\pm3.4}$ | $\underline{59.3}_{\pm2.0}$ | $34.3_{\pm0.6}$ |
| GPRGNN | $O(Km)$ | $\mathbf{83.2}_{\pm0.9}$ | $65.1_{\pm3.1}$ | $63.9_{\pm2.0}$ | $41.5_{\pm1.5}$ | $\underline{35.5}_{\pm0.6}$ |
| JacobiConv | $O(Km)$ | $\underline{82.3}_{\pm0.6}$ | $69.2_{\pm7.1}$ | $73.3_{\pm3.4}$ | $\mathbf{60.5}_{\pm2.6}$ | $34.9_{\pm0.5}$ |
| **Ours** | $O(m)$ | $79.3_{\pm0.4}$ | $\underline{78.4}_{\pm2.4}$ | $\mathbf{80.4}_{\pm1.5}$ | $48.8_{\pm0.9}$ | $\mathbf{35.8}_{\pm0.4}$ |
| $\Delta_{\mathrm{mix}}$ | — | $-1.6$ | $-5.4$ | $\mathbf{+10.6}$ | $-17.1$ | $\mathbf{+6.6}$ |

## 2.4 Permutation Equivariance

**Proposition 1.** *Let* $\mathbf{P}$ *be any* $n{\times}n$ *permutation matrix. Then AdaptiveMixGNN is permutation equivariant:* $f(\mathbf{PX}, \mathbf{PAP}^\top) = \mathbf{P}\,f(\mathbf{X}, \mathbf{A})$.

*Proof.* The $\alpha$-predictor in Eq. equation 2 is a pointwise function of each node representation $\mathbf{h}_i^{(\ell)}$ (row of $\mathbf{H}^{(\ell)}$). Under permutation $\pi$: $\mathbf{H}'^{(\ell)}{=}\mathbf{PH}^{(\ell)}$, so $\alpha'^{(\ell)}_{\pi(i)} = \sigma\left( \left(\mathbf{h}_i^{(\ell)}\right)^\top \boldsymbol{\theta}^{(\ell)} + b^{(\ell)} \right) = \alpha_i^{(\ell)}$, hence $\mathrm{diag}(\boldsymbol{\alpha}'^{(\ell)}) = \mathbf{P}\,\mathrm{diag}(\boldsymbol{\alpha}^{(\ell)})\mathbf{P}^\top$. Since $\tilde{\mathbf{A}}'{=}\mathbf{P}\tilde{\mathbf{A}}\mathbf{P}^\top$ preserves degree, $\mathbf{S}'_{\mathrm{LP}}{=}\mathbf{P}\mathbf{S}_{\mathrm{LP}}\mathbf{P}^\top$ and likewise for $\mathbf{S}_{\mathrm{HP}}$. The adaptive operator transforms as:

$$\mathbf{S}'_{\boldsymbol{\alpha}'}\,\mathbf{X}' = \mathbf{P}\,\mathrm{diag}(\boldsymbol{\alpha})\underbrace{\mathbf{P}^\top\mathbf{P}}_{\mathbf{I}}\mathbf{S}_{\mathrm{LP}}\underbrace{\mathbf{P}^\top\mathbf{P}}_{\mathbf{I}}\mathbf{X} + \dots$$
$$= \mathbf{P}(\mathrm{diag}(\boldsymbol{\alpha})\,\mathbf{S}_{\mathrm{LP}}\,\mathbf{X}) + \dots = \mathbf{P}(\mathbf{S}_{\boldsymbol{\alpha}}\mathbf{X})$$

Since $\mathbf{W}^{(\ell)}$ acts row-wise and $\sigma_{\mathrm{act}}$ is pointwise, equivariance propagates through all layers. This follows the formal criterion of Han et al. (2025): node-wise operations depending only on local features commute with any node relabelling. □

## 3 Experiments

### 3.1 Setup

**Datasets.** We evaluate on Cora (homophilic citation graph, $h{=}0.81$), Texas ($h{=}0.11$) and Wisconsin ($h{=}0.21$) (heterophilic university web-graphs from WebKB, where nodes are web pages and edges are hyperlinks), Chameleon ($h{=}0.23$), and Actor ($h{=}0.22$), using a widely used 60/20/20 random split protocol as in prior work (Lim et al., 2021).

**Baselines.** (1) MLP (no structure); (2) GCN, GAT (homophilic GNNs); (3) H2GCN (Zhu et al., 2020), GPRGNN (Chien et al., 2021), JacobiConv (Wang & Zhang, 2022) (heterophily-specialised). All baselines are $O(|\mathcal{E}|)$ except GPRGNN and JacobiConv ($O(K|\mathcal{E}|)$).

**Training.** The task is multi-class node classification. All parameters—weight matrices $\mathbf{W}^{(\ell)}$ and per-layer gate parameters $(\boldsymbol{\theta}^{(\ell)}, b^{(\ell)})$—are trained jointly via Adam.

**Sanity checks.** Following the evaluation recommendations of Platonov et al. (2023), we validate against potential leakage and spurious structure (Appendix A).

### 3.2 Main Results

Table 1 shows that AdaptiveMixGNN achieves the highest accuracy on Wisconsin (**80.4%**) and competitive results on Texas, outperforming polynomial filters on these strongly heterophilic graphs. On Cora, $\alpha_i$ converges to LP-favouring values ($\bar{\alpha}{=}0.83$), demonstrating graceful adaptation to homophilic structure. On Actor, adaptivity yields **+6.6%** over the best fixed filter.

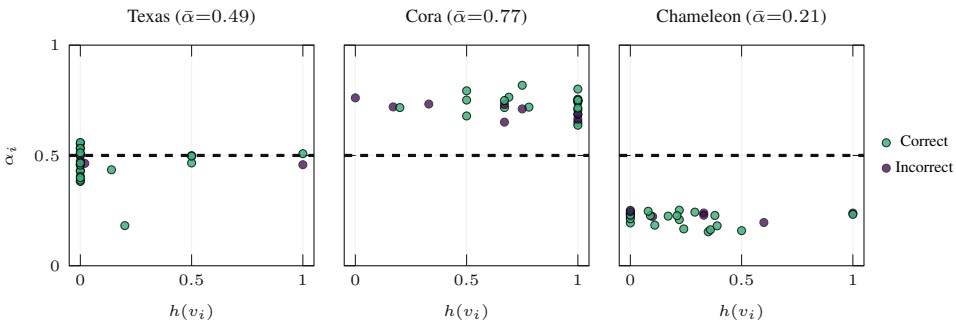

Figure 2: Learned $\alpha_i$ vs. local homophily $h(v_i)$. Panels share the same axes and styling; the $y$-axis is shown once (left) to reduce clutter. Dashed line marks $\alpha_i{=}0.5$ (balanced mix).

**Chameleon as boundary case.** Our model reaches 48.8%, behind JacobiConv (60.5%). Notably, $\Delta_{\mathrm{mix}}{=}{-}17.1\%$: LP-only achieves 65.8%, indicating that Chameleon's community structure benefits from pure smoothing despite low global homophily (Luan et al., 2022). This marks the expressivity limit of first-order adaptive filters.

### 3.3    ABLATION: ADAPTIVITY AS INSURANCE

Ablation experiments (row $\Delta_{\mathrm{mix}}$ in Table 1) expose fundamental asymmetries: on Cora, HP-only collapses to 24.2%; on Texas, LP-only degrades to 57.8%. The optimal fixed filter is *dataset-dependent* and unknowable a priori.

Adaptive $\alpha$ acts as **insurance**: it avoids catastrophic modes and, on Wisconsin and Actor, *exceeds* both fixed baselines (**+10.6%** and **+6.6%** respectively). This confirms that node-wise mixing discovers complementary structure that no uniform filter can capture.

### 3.4    GEOMETRIC GROUNDING OF LEARNED $\alpha$

Figure 2 shows that learned $\alpha_i$ adapts to dataset structure: **Texas** centers near $\bar{\alpha}{=}0.49$ (balanced), **Cora** strongly favours LP ($\bar{\alpha}{=}0.77$), and **Chameleon** shows extreme HP bias ($\bar{\alpha}{=}0.21$). Paradoxically, Chameleon's HP bias *hurts* performance: LP-only achieves 65.8% vs. adaptive's 48.8%. This confirms that Chameleon's community structure benefits from global smoothing despite low node-level homophily—a signature of its boundary-case status where first-order filters cannot disentangle inter-community interference.

## 4    CONCLUSION

AdaptiveMixGNN demonstrates that a *local adaptive inductive bias*—learned per-node mixing between LP and HP graph shift operators—provides a principled middle ground between uniform global filters and complex polynomial architectures. Adding only $d{+}1$ parameters per layer and maintaining $O(|\mathcal{E}|)$ complexity, it achieves the best results among first-order methods on strongly heterophilic graphs, avoids the catastrophic failure modes of fixed filters, and produces geometrically interpretable frequency responses. Chameleon exposes the theoretical limit of single-hop filter banks; future work will explore multi-scale extensions that increase expressivity while preserving linear scaling.

### REPRODUCIBILITY STATEMENT

Code is available at `https://github.com/miguelalcocker/AdaptiveMixGNN`. All results use 60/20/20 splits from Lim et al. (2021) with 10 random seeds.

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

## A  SANITY TESTS

We validate model legitimacy following the evaluation considerations highlighted by Platonov et al. (2023), addressing a key concern for small heterophilic datasets.

**Shuffle Test.** We train with *uniformly random* labels (not permuted, to eliminate class-distribution bias). On Texas with $C=5$ classes: expected accuracy $1/C=20.0\%$; observed: $21.3\%$ ($\leq$ statistical margin $2\sqrt{p(1-p)/N_{\text{test}}}$). The model does not memorise spurious structure.

**Feature Silence.** Replacing all features with **0** degrades Texas accuracy from $78.4\%$ to $52.1\%$ ($-33.6\%$ relative). This confirms that the model uses genuine feature–structure interactions, not topological shortcuts.

**$\alpha$ Dynamics.** The learned $\alpha$ values: (i) deviate from their initialisation at 0.5; (ii) vary across nodes (std$>0.01$); (iii) respond to feature perturbation (different inputs $\Rightarrow$ different $\alpha$). This excludes degenerate convergence to a fixed point.

