# OpenReview forum: "ADAPTIVEMIXGNN: Local Adaptive Inductive Bias for Heterophilic Node Classification"
_ICLR.cc/2026/Workshop/GRaM — ICLR 2026 Workshop GRaM Poster_

### Official Review · Reviewer_SEZY · 2026-02-20

**Rating:** 4
**Confidence:** 3

**Review:**

This paper builds on the observation that most message-passing GNN models act as low-pass filters, which may not be ideal for graphs with varying levels of homophily. It proposes a solution by designing a GNN model, *AdaptiveMixGNN*, that adaptively combines low-pass and high-pass operators on the graph.

**Strengths**
- The introduction and overall manuscript are well-structured.
- There is a sufficient amount of clarifying images and tables.
- The authors propose an efficient and interesting solution after interpreting a limitation of message-passing GNNs through the lens of spectral filters.

**Weaknesses/Questions**
- Parts of the paper are inaccessible to non-experts and require a deeper description of the context and background knowledge. For example: (1) The abstract is quite dense with technical terms, objects, and notations that are not defined beforehand; instead, it should provide a high-level overview of the paper's contributions that is accessible to a broader audience. (2) A similar critique applies to the introduction, which would benefit from providing more background information and being less overly detailed right away (e.g. a non-familiar reader wouldn't know what Texas and Wisconsin represent in this context).
- It appears that the literature has already explored the idea of exploiting graph filters of different frequencies for graph learning ([1][2]), including the adaptive combination of low- and high-pass filters as proposed in this paper (see [1]). What are the similarities and differences between your method and, for example, FAGCN ([1])? I am somewhat confused about the novelty of the work, as these papers are neither mentioned nor discussed in your manuscript. It would be natural to position your work within this already existing context.
- The claim and proof of the permutation equivariance property seem a bit disconnected from your other contributions and experiments. What is the practical purpose of having this property, and how does it play a role in the experiments or in answering your research questions? Given the space constraints, I believe it would be more beneficial to remove this proof and instead dedicate more space to defining important concepts. For instance, providing some background on how GNNs act as low-pass and high-pass filters would be highly valuable, as this is not explained extensively in Sections 2.1 and 2.2.
- How is the training done? Are the adaptive $\alpha$ parameters and the weights $W$ learned simultaneously?
- Reading the experiment setup, some questions are left unanswered: What exactly is the task? How does this experiment answer your research questions, and what did you expect to observe? Why did you choose these specific baselines, and how do they fall into the categories of "low-pass filters" or otherwise?
- The ablation study (the last row of Table 1​) is very unclear in both (1) its definition and (2) its results. Regarding (1), it would be much more explicative to report the raw accuracies corresponding to the LP-only, HP-only, and adaptive $\alpha$ cases directly in the table, rather than just reporting the accuracy gains. This would help the reader understand exactly how the adaptive parameter $\alpha$ plays a role. Regarding (2), the results in gain difference seem inconsistent and confusing, especially the extreme drop in accuracy on Chameleon. Does this suggest that the training procedure for the different parameters suffers from optimization issues? This ties back to why I suggested discussing the training procedure in more detail.
- Still regarding experiments, I would suggest adding a comparison to the previously mentioned line of related works (e.g. [1], [2]).
- Table 1 would be clearer if the homophily levels for each dataset were reported directly in the table header.

**References**

[1] Bo, Deyu, et al. "Beyond low-frequency information in graph convolutional networks." Proceedings of the AAAI conference on artificial intelligence. Vol. 35. No. 5. 2021.

[2] Zhang, Qi, et al. "Beyond low-pass filtering on large-scale graphs via adaptive filtering graph neural networks." Neural Networks 169 (2024): 1-10.

**Pmlr Suitability:**

NA

---

### Official Review · Reviewer_hPiC · 2026-02-23

**Rating:** 5
**Confidence:** 3

**Review:**

## Summary

This paper proposes AdaptiveMixGNN, a first-order spectral graph neural network that introduces node-wise adaptive mixing between low-pass and high-pass graph shift operators. Instead of committing to a single global filter, the model learns a per-node gating coefficient that blends smoothing and sharpening operations. The approach maintains linear-time complexity and introduces only minimal parameter overhead per layer.

The main goal is to provide a lightweight alternative to higher-order polynomial filters or multi-expert architectures when handling graphs with mixed homophilic and heterophilic regimes. Empirical results on several benchmark datasets show competitive performance among first-order methods, particularly on strongly heterophilic graphs.

------

## Strengths

- The idea of allowing each node to interpolate between low-pass and high-pass operators is conceptually straightforward and practically appealing.
- Results on both homophilic and heterophilic datasets illustrate how the learned mixing coefficients adapt to structural differences.
- Adding only d+1 parameters per layer keeps the model lightweight relative to mixture-of-experts approaches.

------

## Weaknesses

- While the idea is clean, the manuscript does not clearly differentiate this approach from prior adaptive filtering or frequency-mixing GNN models. A sharper positioning would help clarify what is genuinely new beyond a simplified formulation.
- The evaluation focuses on relatively small heterophilic datasets. It remains unclear whether the proposed local mixing mechanism retains advantages on larger or more complex graphs.
- Although the method is framed in spectral terms, the geometric implications are not deeply developed.

**Pmlr Suitability:**

NA

---

### Official Review · Reviewer_gczj · 2026-02-24
**A simple and interesting local adaptivity idea**

**Rating:** 5
**Confidence:** 4

**Review:**

This paper proposes a clean, lightweight per-node LP/HP mixing mechanism for heterophilic node classification with essentially GCN-like complexity, and the idea is genuinely appealing in its simplicity, but the empirical case is too limited and the framing overreaches. The results are mixed rather than clearly convincing: the method does well on Texas/Wisconsin and shows useful ablation behavior, but it underperforms badly on Chameleon (including a negative ∆mix) and is not consistently strong across datasets, which weakens the broader claim that local adaptivity robustly solves the global-filter problem. The paper also leans heavily on small heterophilic benchmarks where variance and evaluation choices are known concerns, and while it includes sanity checks, the evidence still feels insufficient to support stronger “best among first-order methods” or general architectural conclusions. Overall, this reads like a promising workshop/tiny-paper idea with good intuition and nice efficiency properties, but not yet a strong or broadly validated advance.

**Pmlr Suitability:**

NA

---

### Meta-Review · Area_Chair_7XFg · 2026-02-24

**Decision:**

Accept

**Metareview:**

This paper introduces AdaptiveMixGNN to learn low-pass and high-pass filters for heterophilic node classification. The reviewers all agreed that the paper is interesting, while having some concerns. I recommend acceptance and ask the author to incorporate the reviewers' comments in the final version of their paper (specifically Reviewer SEZY).

**Relevance To Proceedings:**

Tiny paper — does not apply

**Relevance To Workshop:**

Yes — suitable for GRaM

---

### Decision · Program_Chairs · 2026-03-02

Accept (Poster)